# *Sporothrix* spp. Biofilms Impact in the Zoonotic Transmission Route: Feline Claws Associated Biofilms, Itraconazole Tolerance, and Potential Repurposing for Miltefosine

**DOI:** 10.3390/pathogens11020206

**Published:** 2022-02-03

**Authors:** Giulia Maria Pires dos Santos, Luana Pereira Borba-Santos, Taissa Vila, Isabella Dib Ferreira Gremião, Sandro Antonio Pereira, Wanderley De Souza, Sonia Rozental

**Affiliations:** 1Programa de Biologia Celular e Parasitologia, Instituto de Biofísica Carlos Chagas Filho, Universidade Federal do Rio de Janeiro, Rio de Janeiro 21941-902, RJ, Brazil; giuliafreitas@biof.ufrj.br (G.M.P.d.S.); tvila@biof.ufrj.br (T.V.); wsouza@biof.ufrj.br (W.D.S.); rozental@biof.ufrj.br (S.R.); 2Instituto Nacional de Infectologia Evandro Chagas, Fundação Oswaldo Cruz, Rio de Janeiro 21040-360, RJ, Brazil; isabella.dib@ini.fiocruz.br (I.D.F.G.); sandro.pereira@ini.fiocruz.br (S.A.P.); 3Centro Nacional de Biologia Estrutural e Bioimagem (CENABIO), Cidade Universitária, Rio de Janeiro 21941-902, RJ, Brazil

**Keywords:** biofilms, cat claws, miltefosine, *Sporothrix brasiliensis*, *Sporothrix schenckii*

## Abstract

Sporotrichosis is the most prevalent subcutaneous mycosis globally, and it is typically caused by direct inoculation of the soil saprophytic fungus *Sporothrix* spp. into the patients’ skin. However, sporotrichosis has an important zoonotic transmission route between cats and humans in hot-spot endemic areas such as Brazil. Antifungal itraconazole is the first-line treatment; however, it is frequently associated with recurrence after withdrawal, mainly on cats. Biofilms are important resistance structures related to the environmental persistence of most microorganisms. In the present work, we evaluated *Sporothrix* yeasts’ ability to form biofilms in an ex vivo model of infected claws of cats. Using scanning electron microscopy, we demonstrated the presence of fungal biofilms in the claws of cats diagnosed with sporotrichosis confirmed by isolation of *Sporothrix* spp. in culture. We present here evidence of antibiofilm activity of miltefosine and suggest its use off-label as an antifungal as a putative alternative to itraconazole against *Sporothrix* biofilms. Claw contamination could sustain infections through a continuous inoculation cycle between open lesions and cat claws. Our results further support the off-label use of miltefosine as a promising alternative, especially for mycosis refractory to conventional treatment.

## 1. Introduction

Sporotrichosis is a subacute or chronic infection caused by thermodimorphic fungi of the genus *Sporothrix*. It is a cosmopolitan disease affecting humans and other mammals, and the incidence is increasing worldwide, mainly in tropical and subtropical regions. Sporotrichosis is considered the most prevalent subcutaneous mycosis globally, and Brazil has the highest incidence [1]. 

*Sporothrix schenckii* is the species that is more prevalent worldwide and it is usually associated with soil-acquired infections. In Brazil, sporotrichosis has an important zoonotic transmission route established between cats and humans and, in contrast with the classic route, the zoonotic infection is mainly caused by the species *Sporothrix brasiliensis*. Cats actively participate in the transmission of the disease, with horizontal (cat–cat) and/or vertical (cat–human, cat–dog) transmission of *Sporothrix* species [2,3,4]. 

The ability of *Sporothrix* species to form biofilms from the filamentous saprophytic phase has been previously described and is associated with increased resistance to antifungal agents [5]. Previous work described the isolation of *Sporothrix* spp. from cats’ claws with sporotrichosis either by cultivating claw clippings or by pressing unwashed claws directly onto the surface of culture media. These authors suggested that fungal growth over cats’ claws may be an important way of transmitting *Sporothrix* spp. by traumatic inoculation; however, the in situ biofilm formation has not been shown yet [6,7,8]. 

*Sporothrix* dimorphism is characterized by the presence of the filamentous form during the saprophytic stage at 25 °C and developing yeast cells (parasitic phase) in the mammal host. This temperature-induced transition is an essential morphological adaptation for colonization in mammals and the establishment of infection [2]. Yeast cells are the main infective propagule transmitted by cats; therefore, studying this parasitic phase as starting inoculum of biofilms is vital to understand its importance to zoonotic transmission and endogenous infections in cats. 

Miltefosine is currently used to treat human and canine leishmaniasis in Brazil [9,10]. Over the last few decades, several in vitro and in vivo studies have shown that miltefosine exerts a potent effect against many fungal species, including dimorphic fungi such as *Histoplasma capsulatum*, *Paracoccidioides* spp., *Coccidioides posadassi*, and *Sporothrix* spp. [11,12,13,14]. Thus, this present work aims to (*i*) evaluate the ability of *S. brasiliensis* and *S. schenckii* to form biofilms over cat claw fragments from the yeast (pathogenic) phase of the fungi; (*ii*) investigate whether *Sporothrix* biofilms are naturally present in the claws of cats diagnosed with sporotrichosis; and (*iii*) establish an ex vivo model to evaluate the susceptibility of claws-formed biofilms of *Sporothrix* spp. to itraconazole and miltefosine.

## 2. Results

### 2.1. Sporothrix brasiliensis and S. schenckii Yeasts Develop Mature Biofilms In Vitro

Fresh inoculums of *Sporothrix* spp. yeasts were used to grow biofilms in 96-well microplates for 24, 48, 72, and 120 h. The growth curve for *S. brasiliensis* and *S. schenckii* biofilms showed consistent time-dependent growth of cells (Figure 1A) and total biomass (Figure 1B). Biofilm biomass reached a plateau after 72 h for all strains, while cell metabolic activity continued to increase until 120 h (Figure 1). A fully mature, densely packed biofilm composed exclusively of yeast cells was observed by confocal laser scanning microscopy (CLSM) after 72 h (Figure 2). 

In order to evaluate the impact of nutrient availability on biofilm growth, two different media were used, one nutrient-rich (RPMI) and one nutrient-poor (YNB). Growth kinetics show significantly reduced metabolic activity for both *S. brasiliensis* and *S. schenckii* species in YNB nutrient-poor media compared to RPMI nutrient-rich media (Figure 3A,B). When the biofilm was formed over sterile claws from healthy cats, no statistical difference was observed between biofilms formed in nutrient-rich or nutrient-poor media (Figure 3C).

### 2.2. Electron Microscopy Evaluation of Ex Vivo Biofilms from S. brasiliensis and S. schenckii Grown over Sterilized Cats’ Claws

We evaluated the ability of yeasts of *S. brasiliensis* and *S. schenckii* to form biofilms over sterilized cats’ claws in an ex vivo model of infected claws. SEM images showed that thick biofilms were formed in both growth conditions (YNB or RPMI) by *S. brasiliensis* and *S. schenckii* with abundant extracellular material (Figure 4 and Figure 5, respectively). Biofilms were most frequently observed in the inner space of the center grooves that extend from the top (closer to the quick) to the third quarter of claws (arrows in Figure 4 and Figure 5). Overall, the *S. schenckii* strain showed a thicker biofilm on rich media than the *S. brasiliensis* strains, while similar biofilm density was observed for both species grown in poor media (Figure 4 and Figure 5).

### 2.3. Cat Claws Powder Contributes to S. brasiliensis and S. schenckii Yeast Cells Growth 

To confirm that *Sporothrix* cells can use claws as a nutrient source, yeasts of *S. brasiliensis* and *S. schenckii* were incubated in YNB medium (poor medium) supplemented with a suspension containing 2% of cat claws powder for 24, 48, 72, and 120 h. As illustrated by the metabolic activity ratio in Figure 6, all *Sporothrix* strains exhibited higher growth in the YNB medium enriched with powdered claws than in YNB alone after all incubation times (values higher than 1), except the Ss 245 strain at 120 h.

### 2.4. Fungal Biofilms Were Identified in the Claws of Cats Clinically Diagnosed with Sporotrichosis

To confirm that cat claws could harbor *Sporothrix* spp. biofilms, claws from cats diagnosed with sporotrichosis were analyzed by SEM. Ten samples were collected from a total of 10 animals (clinical information is described in Table 1), and fungal biofilms were identified in 7 samples, from 6 different cats (70% and 66.6% positive-rate, respectively) (Figure 7). Biofilms were mostly found in the center grooves, closer to the immediate area. Biofilm-positive samples presented deep-seated biofilms, growing inside the grooves and caves of the nail, densely covered in extracellular material and actively penetrating through the nail cavities. 

### 2.5. Sporothrix spp. Yeast Cells Were Susceptible to Commercial Antifungals and Miltefosine

Planktonically grown yeasts from all strains tested were susceptible to itraconazole, amphotericin B, and miltefosine (Appendix A). Miltefosine showed great inhibitory activity and a potential fungicidal profile (Appendix A). 

### 2.6. In Vitro Sporothrix spp. Biofilms Showed Low Susceptibility to Itraconazole but Were Susceptible to Amphotericin B and Miltefosine 

120-h-old *S. brasiliensis* and *S. schenckii* biofilms were susceptible to amphotericin B, with MIC_80_ of approximately 2 µg/mL (Figure 8B), but showed reduced susceptibility to itraconazole (Figure 8A). Concentrations between 4−8 µg/mL of miltefosine eliminated 50% and 80% of *S. brasiliensis* and *S. schenckii* biofilms, respectively (Figure 8C).

### 2.7. Ex Vivo Sporothrix spp. Biofilms Were Susceptible to Miltefosine

Ex vivo *Sporothrix* biofilms grown for 120 h over sterile cat claws were exposed to the MIC_80_ of antifungals for 48 h. *S. brasiliensis* biofilms showed mild reduction (*p* < 0.01) after treatment with high concentrations of itraconazole (64 µg/mL) (Figure 9A), while the same concentration did not affect *S. schenckii* biofilm viability (Figure 9B). Noteworthily, treatment with 8 µg/mL of miltefosine resulted in a drastic reduction in biofilm viability of both *S. brasiliensis* (*p* < 0.001) and *S. schenckii* (*p* < 0.01) (Figure 9). *S. brasiliensis* biofilms treated with itraconazole and miltefosine were also analyzed by SEM, revealing that miltefosine induced more morphological changes than itraconazole in fungal cells (Figure 10).

## 3. Discussion

In Brazil, sporotrichosis occurs as sapronoses (e.g., soil–human), zoonosis (e.g., cat–human), or horizontal animal transmission (e.g., cat–cat, cat–dog). Cat scratches play an important role in the epidemiologic chain of transmission of *Sporothrix* sp. [15]. The report by Schubach et al., 2001 was the first to describe the isolation of *S. schenckii* directly from the claw surfaces of infected cats through the incubation of claw clippings in a culture medium [6]. Later on, Souza et al., 2006 and Borges et al., 2013 were able to isolate this fungus by pressing unwashed claws of the forelimbs of infected cats directly onto the surface of mycobiotic agar medium [7,8]. More recently, Brilhante et al., 2021 described an ex vivo model of biofilm using the filamentous phase of *Sporothrix* species and cat claws [16].

We report the *Sporothrix* biofilm formation using the yeast phase as the starting inoculum. During its life cycle, *Sporothrix* sp. grows as yeast during the infective phase and is also described as “the pathogenic form”. Since infected cats harbor mainly yeasts in their open wounds, the underlying characteristics of biofilms developed starting from yeasts can contribute to a better understanding of the pathophysiology of the disease, while also highlighting the importance of treating the cats with sporotrichosis as an important measure of disease control, as it induces a quick reduction on the fungal burden [17]. 

We observed that *S. schenckii* and *S. brasiliensis* exhibited different growth kinetics during biofilm formation, with a higher metabolic activity of *S. schenckii* than *S. brasiliensis* during the first time of incubation. We include in the study only one representative isolate of *S. schenckii* because the main species related to animal sporotrichosis in Brazil is *S. brasiliensis* [1], although *S. schenckii* could also cause sporotrichosis in cats [18]. Besides different growth kinetics, *S. schenckii* and *S. brasiliensis* also exhibited a distinct susceptibility profile, *S. schenckii* being less susceptible to antifungals in vitro than *S. brasiliensis*, as reported by our group previously [19].

The kinetics of fungal biofilm formation presents a significant difference when using a poor (YNB) or a rich (RPMI 1640 supplemented with 2% glucose) medium; however, this difference disappears when the claws are added to the system. This is corroborated by the higher growth of both species when poor media was supplemented with sterile powdered claws. Furthermore, the presence of cat claws also favored biofilme growth, as seen by SEM (Figure 4 and Figure 5). While in vitro biofilms grew much better in rich culture media compared to poor culture media, for ex vivo biofilms no difference in biofilm thickness or extracellular matrix formation was observed. Taken together, these results indicate that *S. brasiliensis* and *S. schenckii* can use cat claws as a source of the nutrient.

Previous studies from Moharram and Abdel-gawad (1989) associated keratolytic fungal activity and the growth over rabbit claws [20]. In 2003, Sharma and Rajak described the importance of keratinolytic fungi in nature, whose biological function in the soil is the degradation of keratinized materials, such as hides, skins, claws, and nails, among other kenatinized materials [21]. 

Keratinolytic activity through enzyme secretion is a known virulence factor of several saprophytic fungi; however, it has never been demonstrated in *Sporothrix* spp. Recently, Prakash et al., 2020 identified 250 genes encoding predicted proteases in the genome of *Sporothrix* species. The most abundant were serine proteases, such as prolyl oligopeptidase, which play a role in extracellular degradation and protein maturation, serine carboxypeptidase, and X-Pro dipeptidyl peptidase [22]. Another notorious family of proteases present in *Sporothrix* species is the eukaryotic aspartyl protease, which in *Candida albicans* plays a role in the degradation of host tissues, and in immune evasion by the degradation of proteins involved in the immune response [23]. Further studies are needed to fully characterize the proteolytic arsenal of *Sporothrix* species and to determine whether keratinolytic-like activity is one of its virulence weapons. 

To further translate the observed phenomena to the clinical setting, we analyzed claws from cats previously diagnosed with sporotrichosis using a powerful electron microscopy technique and confirmed that 70% of the cat’s claws analyzed harbored fungal biofilms with hyphae and yeast-like structures. It is known that biofilms are biological communities with a high degree of organization, in which microorganisms form structured, coordinated, and functional communities. These biological communities are incorporated into a self-created extracellular matrix and are associated with a high level of antimicrobial resistance of associated organisms [5]. Biofilm formation in the claws of infected cats could set up an infectious reservoir and perpetuate the transmission of resistant yeasts in the cat–cat and cat–human routes, increasing the frequency of isolates unresponsive to available treatments. Therefore, the identification of drugs that can effectively eliminate fungal biofilms is of the utmost importance.

Miltefosine, an alkylphosphocholine used as an antitumor and antiparasitic drug, is a strong candidate for repositioning of drugs. Previous works have already shown its antifungal efficacy against *Candida*
*species*, *Cryptococcus*, *Sporothrix*, *Paracoccidioides*, and several other fungi [11,12,13,14,24,25]. The fact that concentrations up to 8 µg/mL of miltefosine were able to inhibit *Sporothrix* biofilms is a good result, considering that several compounds that inhibit planktonic fungal cells do not inhibit biofilms because biofilms were less susceptible to compounds due to cell density, extracellular matrix, and gene expression alterations [26]. The mechanism of action related to the antifungal activity of miltefosine is not fully understood; however, it is partly due to the induction of apoptotic-like cell death and increased plasma membrane permeability [27]. Spadari et al., 2018 and Rollin-Pinheiro et al., 2021 showed that miltefosine decreased mitochondrial membrane potential and increased ROS levels in *Cryptococcus neoformans* and *Scedosporium aurantiacum* [24,28]. Miltefosine induces pronounced disturbances in the plasma membrane in planktonic *Sporothrix* cells [13]. We hypothesized that miltefosine acts on *Sporothrix* biofilms by disrupting the cell membrane integrity, thus promoting an electrolyte imbalance and consequently the cell death of the fungus.

The data presented here corroborate the growing literature supporting repositioning miltefosine as a potent antifungal drug. Collectively, miltefosine was effective against both planktonic and biofilms of *S. brasiliensis* and *S. schenckii*, with greater inhibition of the biofilm formation than itraconazole, the drug of choice for the treatment of sporotrichosis.

Silva et al., 2018 evaluated the effectiveness and safety of the administration of miltefosine in a small cohort of 10 cats with resistant sporotrichosis [29]. All cats were previously treated for more than one year with itraconazole at a dosage of 100 mg/kg/day and recognized as refractory to treatment. Between the 10 cats studied, only one received the treatment with miltefosine (2 mg/kg orally every 24 h) for 45 days; the other six received it for 30 days. Although miltefosine did not cause adverse/toxic events in the studied cats, it did not lead to clinical remission of refractory sporotrichosis to conventional treatments in these patients [29].

In conclusion, we showed that the yeast phase of *S. brasiliensis* and *S. schenckii* isolates can form mature, robust biofilms on the claws of cats. We further demonstrated that *Sporothrix* sp. is able to use claw components as a nutrient source. Finally, we confirmed a high frequency of fungal biofilms in the claws of cats with sporotrichosis, which may represent one of the fundamental factors for the transmission of *Sporothrix* spp. In addition, we demonstrated that miltefosine is effective against *Sporothrix* biofilms formed in an ex vivo model using cat claws. Further studies are needed to investigate the effectiveness and safety of this drug in vivo in cats with sporotrichosis.

## 4. Materials and Methods

### 4.1. Strains and Growth Conditions 

The reference strains *S. brasiliensis* ATCC 4823, *S. brasiliensis* ATCC 4824, *S. schenckii* ATCC 32286, and the clinical isolate *S. brasiliensis* ss245 were used in this work. The clinical isolate *S. brasiliensis* ss245 [30] was isolated from skin lesions of a cat at Instituto Nacional de Infectologia Evandro Chagas (INI), Fundação Oswaldo Cruz (Fiocruz), Rio de Janeiro, Brazil. All isolates were maintained in 20% glycerol stocks at –20 °C. A small inoculum from stocks was propagated weekly in YPD liquid medium (10% yeast extract, 20% peptone, 30% dextrose) pH 4.5 at room temperature, under constant shaking, for 7 days (filamentous form). In order to obtain the yeast form, a small inoculum (50−100 μL) from the filamentous growth was inoculated into YPD liquid medium pH 7.8 and incubated at 36 °C for 7 days under constant shaking [31]. 

### 4.2. Feline Claws Samples 

Claws were collected between years 2020 and 2021 from ten cats with sporotrichosis confirmed by isolation of *Sporothrix* spp. in culture. The cats were assisted at the Laboratório de Pesquisa Clínica em Dermatozoonoses em Animais Domésticos (Lapclin-Dermzoo (INI/Fiocruz). Experiments were conducted after the owner signed an informed consent form. All procedures were approved by the Animal Ethics Committee (CEUA-Fiocruz) under the license number LW−11/21. Claws from healthy cats were obtained from the discarded material of Lapclin-Dermzoo, (INI, Fiocruz) as they are clipped during consultations. 

### 4.3. In Vitro and Ex Vivo Biofilm Formation

To induce biofilm formation, yeast suspensions were washed once in growth media, resuspended in RPMI 1640 supplemented with 2% of glucose (nutrient-rich media; Sigma, Ronkonkoma, NY, USA) or YNB (nutrient-poor media; BD Difco, Baltimore, MD, USA) to a final concentration of 1 × 10^6^ cells/mL and plated (100 µL) on 96-well polystyrene microplates. Plates were incubated for 24, 48, 72, or 120 h at 35 °C under 5% CO_2_. For ex vivo biofilm formation, sterile claws collected from healthy cats and previously autoclaved were placed inside the microplates wells, and a yeast suspension in RPMI 1640 supplemented with 2% of glucose or YNB of concentration 1 × 10^6^ cells/mL was plated (200 µL) in the wells containing the claws. The plates were incubated for 120 h at 35 °C in 5% CO_2_.

### 4.4. Biofilm Quantification

Biofilm biomass and metabolic activity of the cells were quantified by crystal violet staining assay and the XTT-reaction assay, respectively, as previously described [32,33]. For both assays, the spectrophotometric reading was carried out using EMax Plus (Molecular Devices, San Jose, CA, USA) at 492 nm (XTT assay) or 570 nm (for crystal violet assay). Ex vivo biofilms were quantified by colony-forming units (CFU). For that, claws containing biofilms were collected into a sterile tube containing 1mL of RPMI and sonicated for 10 min to disrupt the biofilms. After intense vortexing (30 s, maximum speed), samples were diluted in the same media, plated in BHI agar, and incubated for 4–6 days at 35 °C for CFU enumeration. 

### 4.5. Planktonic Yeast Growth Assay in Powdered Claw Suspension 

Healthy cat claws were crushed, autoclaved, and subsequently mixed with YNB medium to form a suspension with 2% powdered claws. An inoculum of 10^6^ cells/mL was prepared in YNB medium with or without (control) powdered claws and added to the well of a 96-well plate (200 µL/well). Plates were incubated at 35 °C with 5% CO_2_ for 24, 48, 72, or 120 h. After each timepoint, cell metabolic activity was quantified by XTT-reaction assay.

### 4.6. Planktonic Yeast Susceptibility Assays

Minimal inhibitory concentration (MIC) of antifungals was determined for *Sporothrix* spp. yeast following microdilution broth protocols with modifications, as previously described by our group [34,35,36]. Itraconazole, amphotericin B (Sigma) and miltefosine (Cayman Chemical Company) were serially diluted in 96-well microplates to a final range of 0.03−16 µg/mL. Yeasts were propagated as described above and standardized to 2 × 10^5^ cells/mL in RPMI supplemented with 2% glucose. A total of 100 µL of cell suspension was added to the microplates containing diluted antifungals and incubated for 48 h at 35 °C with 5% CO_2_. Microplates were visually inspected using an inverted microscope (Zeiss Axiovert 100, Reinsdorf, Germany), and growth was quantified spectrophotometrically at 492nm (EMax Plus, Molecular Devices, San Jose, CA, USA). The minimal inhibitory concentration of 50% and 90% of growth (MIC_50_ and MIC_90_, respectively) was calculated in relation to untreated controls. The minimal fungicidal concentration (MFC) was determined by spotting 10 µL from the wells without visible fungal growth onto fresh BHI agar free of drugs. The lowest drug concentration showing absence of regrowth on BHI free of drugs after 5 days, at 35 °C, 5% CO_2_, was considered the MFC.

### 4.7. Biofilm Susceptibility Assays

Biofilms were grown in 96-well polystyrene plates as described above for 120 h, at 35 °C, under 5% CO_2,_ and then exposed to serially diluted concentrations of itraconazole, amphotericin B or miltefosine ranging from 1−128 µg/mL, for 48 h at 35 °C, 5% CO_2_. Subsequently, the metabolic activity of biofilm cells was quantified using the XTT-assay, and the minimal inhibitory concentration of 50% and 80% (MICB_50_ and MICB_80_, respectively) was calculated in relation to untreated biofilms.

### 4.8. Susceptibility of Ex Vivo Biofilms to Antifungals

Biofilms grown over the claws of healthy cats for 120 h in RPMI 1640 supplemented with 2% glucose were transferred to the adjacent clean well containing 200 µL of antifungals at their ½ MIC_80_ concentrations (itraconazole (32 µg/mL), miltefosine (4 µg/mL)) or fresh media (untreated controls) and incubated at 35 °C under 5% CO_2_ for additional 48 h. The remaining biofilms in the claws were quantified by colony-forming units (CFU) enumeration as described above, and biofilm inhibition was calculated concerning untreated controls.

### 4.9. Confocal Laser Scanning Microscopy

Biofilms were grown in vitro in glass-bottom dishes (Cellvis inc., Houston, TX, USA) as described above for 72 h, fixed in 4% paraformaldehyde solution, then stained with a cocktail mix containing calcofluor white (0.01%) and concanavalin-A conjugated to alexa-fluor 488 or wheat germ agglutinin (WGA) conjugated to alexa-fluor 555 (50 µg/mL for both lectins), for 45 min at 35°C. After that, samples were immediately observed in a spinning disk confocal microscope (Cell Observer Yokogawa, Zeiss, Houston, TXGermany), and z-stacks were processed using Fiji software (ImageJ; NIH, USA).

### 4.10. Scanning Electron Microscopy (SEM)

Biofilms were grown for 120 h in nutrient-rich (RPMI 1640 supplemented with 2% glucose) or nutrient-poor media (YNB) as described above, over sterile claws collected from healthy cats and then processed for SEM. Claws collected from cats clinically diagnosed with sporotrichosis were also processed for SEM following the same protocol. Briefly, samples were fixed in paraformaldehyde (4%) and glutaraldehyde (2.5%) solution, washed in cacodylate 0.1M buffer, post-fixed in osmium tetroxide (2%), and dehydrated in a series of ethanol dilutions (30, 50, 70, 90, and 100%). Samples were critical-point-dried in CO_2_, sputtered with gold, and observed under a high vacuum (15 kV) in a scanning electron microscope (Quanta 250, FEI).

### 4.11. Statistical Analysis

All statistical analysis was performed using GraphPad Prism 8.0 software. Experiments were performed on at least 3 separate occasions and in triplicate where applicable, and averages were used to present data. Multiple groups were compared by ANOVA with Tukey’s multiple-comparison correction, and Student’s unpaired *t*-test was used to compare differences between two samples. For all statistical analyses, *p* values < 0.05 were considered significant. 

## Figures and Tables

**Figure 1 pathogens-11-00206-f001:**
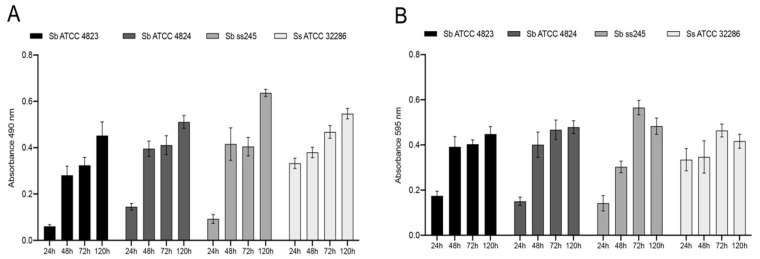
In vitro growth kinetics of *S. brasiliensis* (Sb) and *S. schenckii* (Ss) biofilms in RPMI supplemented with 2% glucose. (**A**) Metabolic activity of biofilm cells quantified by the XTT assay; (**B**) Total biofilm biomass was quantified by the crystal violet assay.

**Figure 2 pathogens-11-00206-f002:**
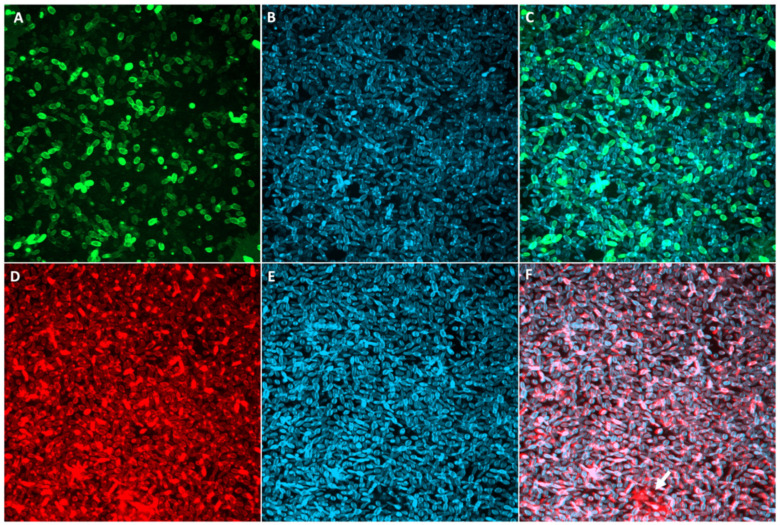
Representative confocal fluorescence images of *S. brasiliensis* ATCC MYA 4823 mature biofilms grown in RPMI + 2% glucose for 72 h at 35 °C. Biofilms stained with concanavalina A (ConA) + FITC (**A**), calcofluor white (**B** and **E**), and wheat germ agglutinin WGA + alexa fluor 546 (**D**). Panels C and F show an overlap of ConA (**C**) or WGA (**F**) and calcofluor. The white arrow points to extracellular matrix accumulation.

**Figure 3 pathogens-11-00206-f003:**
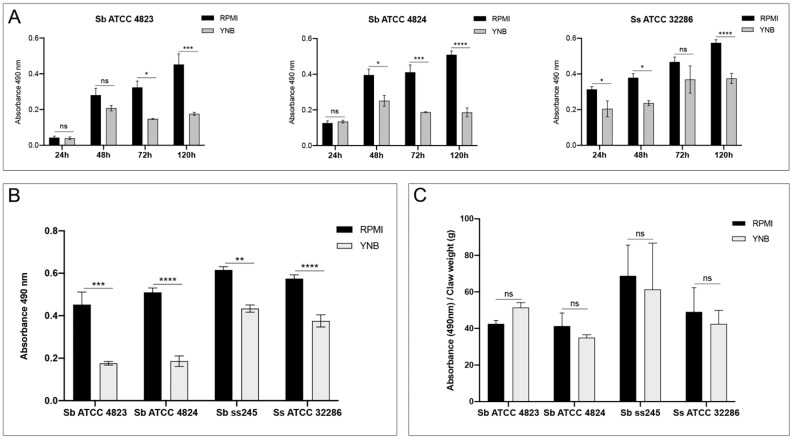
Differences in the biofilm growth using a rich (RPMI) or a poor (YNB) medium. (**A**) Biofilm growth kinetics of *S. brasiliensis* (Sb) and *S. schenckii* (Ss) strains on RPMI + 2% glucose and in YNB media; (**B**) Comparative evaluation of mature biofilms of *Sporothrix* spp. strains grown in RPMI + 2% glucose and YNB media for 120 h; (**C**) Comparative evaluation of ex vivo biofilms grown on sterile claws from healthy cats of *Sporothrix* spp. strains grown in RPMI + 2% glucose and YNB media for 120 h on sterile cat claws. ns, not significant; *, *p* < 0.05; **, *p* < 0.01; ***, *p* < 0.001; ****, *p* < 0.0001.

**Figure 4 pathogens-11-00206-f004:**
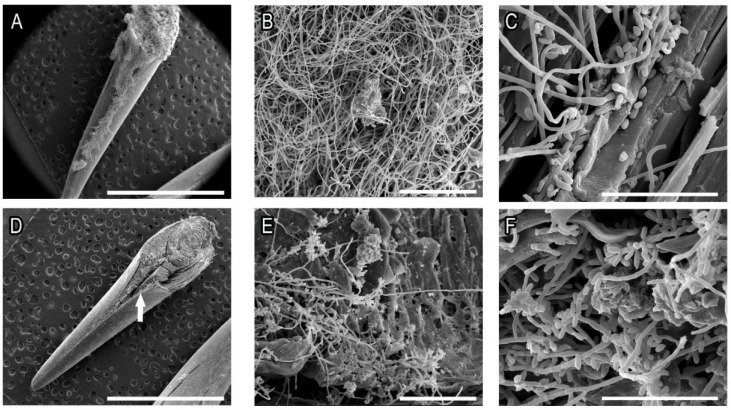
Ex vivo *S. brasiliensis* ATCC MYA 4823 biofilms grown for 120 h in RPMI + 2% glucose (**A**–**C**) or YNB (**D**–**F**) on sterile cat claws. The arrows indicate the area where biofilms were most frequently observed. Bars: (**A**,**D**): 3mm; (**B**,**E**): 50 µm; (**C**,**F**): 30 µm.

**Figure 5 pathogens-11-00206-f005:**
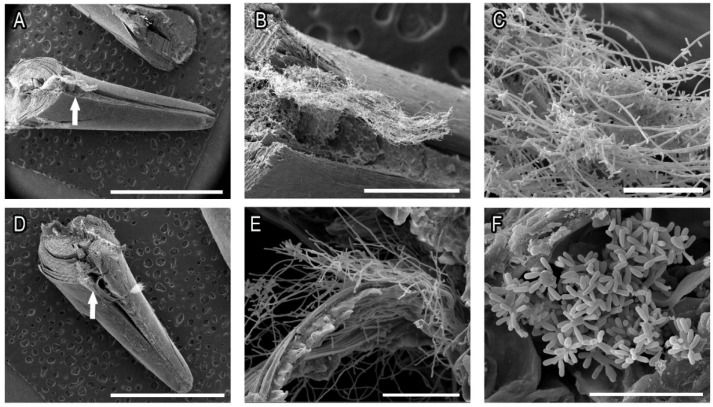
Ex vivo *S. schenckii* ATCC 32286 biofilms grown for 120 h in RPMI + 2% glucose (**A**–**C**) or YNB (**D**–**F**) on sterile cat claws. The arrows indicate the area where biofilms were most frequently observed. Bars: (**A**,**D**): 3mm; (**B**): 500 µm; (**C**, **E**): 50 µm; (**F**): 30 µm.

**Figure 6 pathogens-11-00206-f006:**
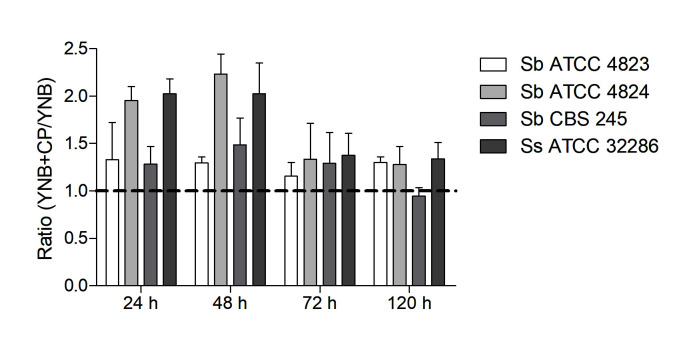
Metabolic activity ratio of *S. brasiliensis* (Sb) and *S. schenckii* (Ss) strains grown in YNB supplemented with 2% claws powder (YNB + CPY) and YNB medium. The XTT assay quantified the metabolic activity of cells.

**Figure 7 pathogens-11-00206-f007:**
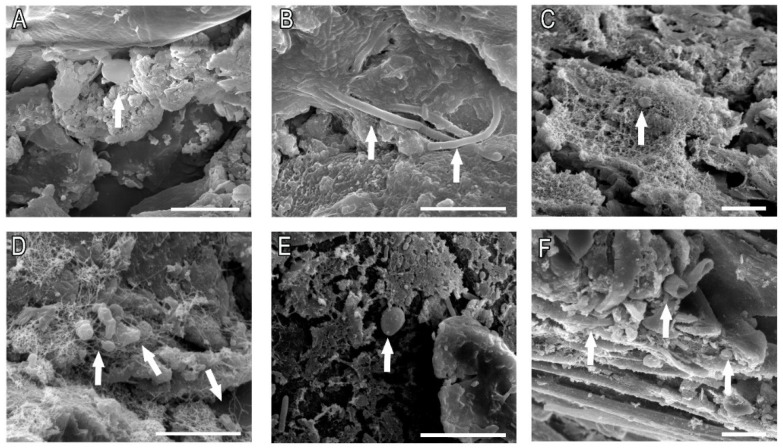
Biofilms on claws recovered from cats diagnosed with sporotrichosis. (**A**) Cat 2; (**B**) Cat 3; (**C**) Cat 6B; (**D**) Cat 8; (**E**) Cat 9; (**F**) Cat 10. The arrows indicate *Sporothrix* cells. Bars: 10 µm.

**Figure 8 pathogens-11-00206-f008:**
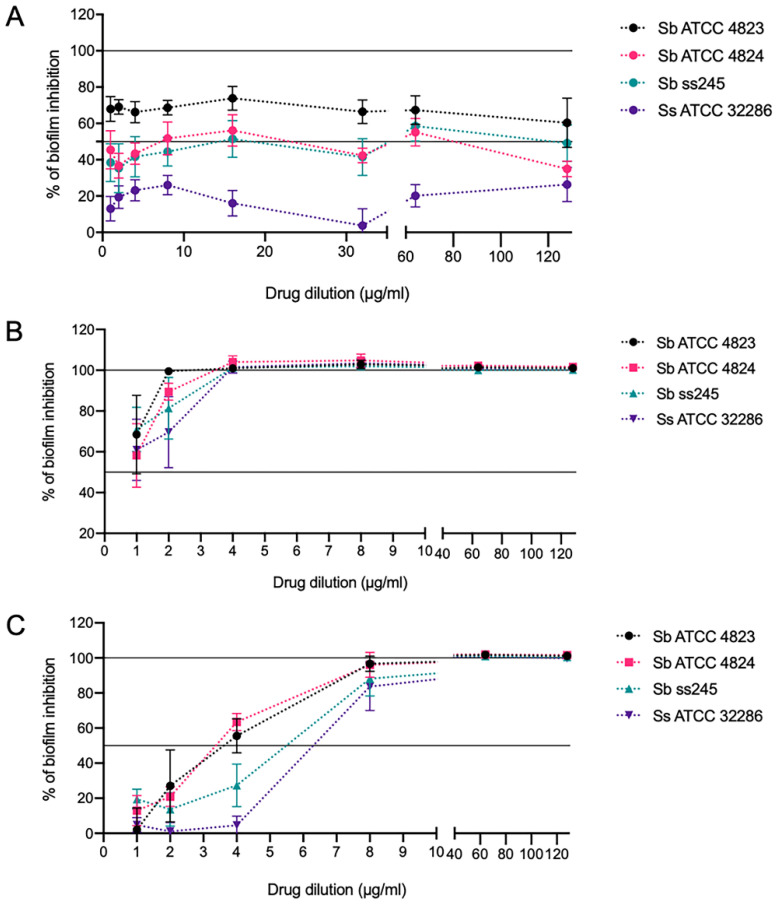
In vitro susceptibility of *Sporothrix* spp. biofilms to (**A**) itraconazole; (**B**) amphotericin B; (**C**) miltefosine. Biofilms grown in 96-well plates for 120 h were exposed to serially diluted drugs for 48 h, and the metabolic activity of biofilm cells after treatment was quantified by the XTT assay. Graphs show % of growth inhibition calculated concerning untreated control wells.

**Figure 9 pathogens-11-00206-f009:**
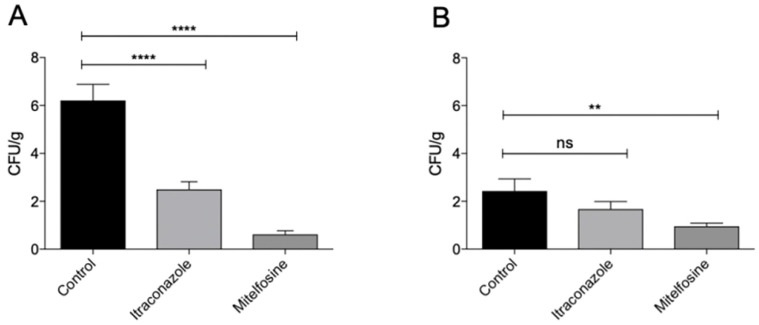
Ex vivo susceptibility of *Sporothrix* spp. biofilms grown on sterile cat claws. Biofilms of *S. brasiliensis* ATCC MYA 4823 (**A**) or *S. schenckii* ATCC 32286 (**B**) were grown for 120 h, and were exposed to the MIC80 of itraconazole (64 µg/mL) or miltefosine (8 µg/mL) for 48 h. Recovered biofilms were plated and the colony-forming units normalized by the biofilm weight. Control biofilms were incubated with fresh media (RPMI 2% glucose). ns, not significant; **, *p* < 0.01; ****, *p* < 0.001.

**Figure 10 pathogens-11-00206-f010:**
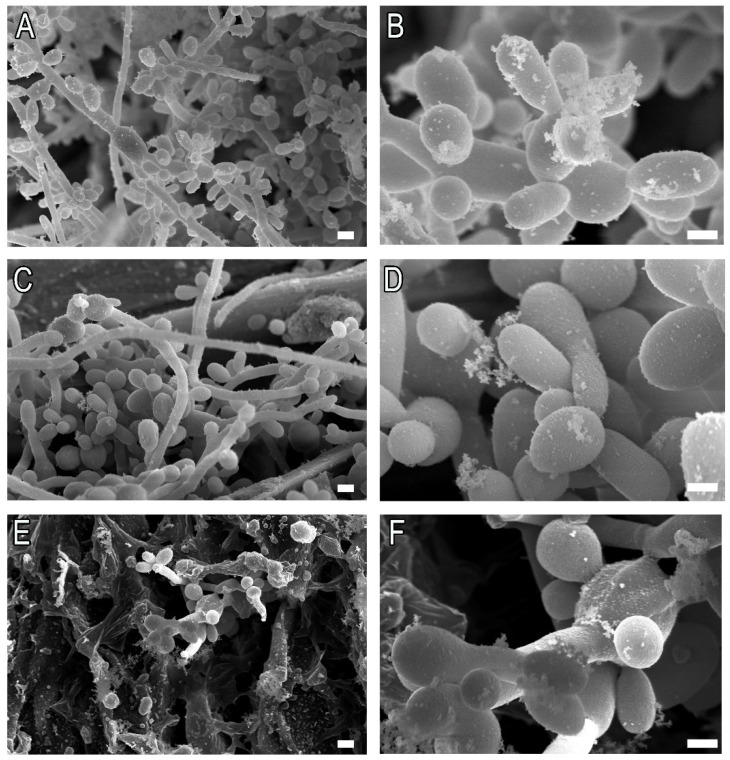
Ex vivo *S. brasiliensis* ATCC MYA 4823 biofilms untreated (**A**,**B**) and treated with 32 µg/mL itraconazole (**C**,**D**) or 4 µg/mL miltefosine (**E**,**F**). Bars: (**A**,**C**,**E**): 2 µm; (**B**,**D**,**F**): 1 µm.

**Table 1 pathogens-11-00206-t001:** Clinical data from cats with a confirmed diagnosis of sporotrichosis. Biofilm presence was considered “yes” (Y) when visible fungal cells were observed. Claws showing bacterial biofilms only were considered “negative” (N).

Cat	Lesions		Therapy at the Moment of Sample Collection **	Previous ITZ Use	Biofilm Positive Claws
	Cutaneous	Mucosal			
1	Y	N	N	N	N
2	Y	N	N	N	Y
3	Y	Y	N	Y	Y
4	Y	Y	N	Y	N
5	Y	Y	N	N	I
6A *	Y	Y	ITZ	Y	Y
6B *	Y	Y	FLZ; TRB	Y	Y
7	N	Y	ITZ; KI	Y	Bacterial biofilm
8	Y	Y	ITZ; KI	Y	Y (mixed biofilm)
9	Y	Y	ITZ; KI	Y	Y (mixed biofilm)
10	Y	Y	N	N	Y

Note:Y = Yes, N = No, I = Inconclusive. * The same animal provided samples in two visits, >2 months apart. A: first collected sample; B: second collected sample; ** itraconazole (ITZ), terbinafine (TRB), potassium iodide (KI).

## Data Availability

The data of this study are available in the presented manuscript.

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
