# Peer review of "Sporothrix spp. Biofilms Impact in the Zoonotic Transmission Route: Feline Claws Associated Biofilms, Itraconazole Tolerance, and Potential Repurposing for Miltefosine"

_pathogens, 2022, doi:10.3390/pathogens11020206_

Round 1

Reviewer 1 Report

The work is well written and provides consistent information on the biological aspects of a fungus of high epidemiological importance. However, some points of the methodology and results were not clear to me.

Starting with the methodology, in lines 343-349, the in vitro and ex vivo biofilm induction protocol is described. The authors state that the sterile claws were placed in the wells of the plate, however, it is not clear whether they are covered by the culture medium containing the yeast suspension or how the  inoculum is performed.

In case the claws are covered by the culture medium, although the results show that the fungus grows better in a medium supplemented with the pulverized claws, a control is needed that demonstrates that the claws are the substrate of this experiment, and not the rich nutrient culture medium. 

Regarding the results, in line 109, the microscopy of only one S. brasiliensis isolate is shown. Was this methodology not applied to the other isolates?

Line 116, fig 3 - To what do you attribute the higher metabolic activity of S. schenckii compared to S. brasiliensis in the first 24h? Why was only one isolate of S. schenckii used?

Lines 140 - 147 - In this paragraph it is not clear whether was evaluated the biofilm formation ability of S. brasiliensis OR of Sb and Ss .

Lines 230-232 - Why were Ss biofilms not evaluated by SEM, if miltefosine showed significant activity against this isolate?

Minor concerns

1 - It is interesting to note that Ss showed lower susceptibility to antifungals tested in vitro, even though Sb is described as the most virulent species.

2 - Please, review the species nomenclature in the legends, as they are not italicized.

Author Response

Dear Reviewer 1,

We are grateful for all suggestions and the revision process that improved our manuscript. The manuscript “pathogens: 1553524” was revised according to the reviewer's comments.

Responses to reviewer 1's comments are described below:

Reviewer #1:

The work is well written and provides consistent information on the biological aspects of a fungus of high epidemiological importance. However, some points of the methodology and results were not clear to me.

ANSWER: We thanked Reviewer 1 for the comments and changed the manuscript considering all suggestions.

Starting with the methodology, in lines 343-349, the in vitro and ex vivo biofilm induction protocol is described. The authors state that the sterile claws were placed in the wells of the plate, however, it is not clear whether they are covered by the culture medium containing the yeast suspension or how the inoculum is performed.

ANSWER: To induce biofilm formation, yeast suspensions were washed once in growth media, resuspended in RPMI 1640 supplemented with 2% of glucose (nutrient-rich media; Sigma, USA) or YNB (nutrient-poor media; BD Difco, USA) to a final concentration of 1x106 cells/ml and plated (100 µl) on 96-well polystyrene microplates. Plates were incubated for 24, 48, 72, or 120hrs at 35 ËšC under 5% CO2. For ex vivo biofilm formation, sterile claws collected from healthy cats and previously autoclaved were placed inside the microplate’s wells. Yeast suspension (1x106 cells/ml) prepared in RPMI 1640 supplemented with 2% of glucose or YNB was plated (200 µl) in the wells containing the claws. The plates were incubated for 120 hours at 35°C in 5% CO2. This information was included in the Materials and Methods (lines 319-327).

In case the claws are covered by the culture medium, although the results show that the fungus grows better in a medium supplemented with the pulverized claws, a control is needed that demonstrates that the claws are the substrate of this experiment, and not the rich nutrient culture medium. 

ANSWER: In Figure 3, we compare the fungal growth in the rich and poor nutrient media (Figure 3B) with growth when claws were added to the system (Figure 3C). These results demonstrated that the claws are related to increased growth, considering that experiments were performed under the same conditions.

Regarding the results, in line 109, the microscopy of only one S. brasiliensis isolate is shown. Was this methodology not applied to the other isolates?

ANSWER: We performed the electron microscopy evaluation of ex vivo biofilms using only one representative reference isolate of each species because the aim was to illustrate the results described in Figure 3C and reach this objective.

Line 116, fig 3 - To what do you attribute the higher metabolic activity of S. schenckii compared to S. brasiliensis in the first 24h? Why was only one isolate of S. schenckii used?

ANSWER: The higher metabolic activity of S. schenckii than S. brasiliensis in the first 24h is due to the different growth kinetics of each species. We include in the study only one representative isolate of S. schenckii because the main species related to animal sporotrichosis in Brazil is S. brasiliensis [doi: 10.3390/jof6040247], although S. schenckii could also cause sporotrichosis in cats [doi: 0.1093/mmy/myz106]. This information was included in the Discussion (lines 218-222).

Lines 140 - 147 - In this paragraph it is not clear whether was evaluated the biofilm formation ability of S. brasiliensis OR of Sb and Ss.

ANSWER: The biofilm formation ability was evaluated using reference isolates of S. brasiliensis and S. schenckii, as described in 2.2 subitem (lines 105-113).

Lines 230-232 - Why were Ss biofilms not evaluated by SEM, if miltefosine showed significant activity against this isolate?

ANSWER: We chose performed SEM with treated cells using one reference isolate (S. brasiliensis ATCC MYA 4823) as a representative model to illustrate miltefosine effects in Sporothrix biofilms.

Minor concerns

1 - It is interesting to note that Ss showed lower susceptibility to antifungals tested in vitro, even though Sb is described as the most virulent species.

ANSWER: S. schenckii and S. brasiliensis exhibited different susceptibility profiles, as previously demonstrated by our group [doi: 10.1371/journal.pone.0240658]. This information was included in the Discussion (lines 222-225).

2 - Please, review the species nomenclature in the legends, as they are not italicized.

ANSWER: The modification was performed.

Reviewer 2 Report

This manuscript number pathogens-1553524 entitled: “Sporothrix spp. biofilms impact in the zoonotic transmission route: feline claws associated biofilms, itraconazole tolerance, and potential repurposing for miltefosine” by Pires dos Santos, et al., demonstrated the presence of Sporothrix fungal biofilms in the nails of cats diagnosed with sporotrichosis and evaluated the anti-biofilm effects of Sporothrix spp of drugs such as Itraconazole, amphotericin B, and miltefosine. The latter was suggested as an alternative to the use of itraconazole against biofilms of the fungus. These results support the use of miltefosine as a promising treatment alternative against sporotrichosis. Eventhough more studies should continue to be carried out for improving the effectiveness and safety in vivo in cats with sporotrichosis The points to review in the document are the following: 1. Modify Sporothrix sp by “Sporothrix spp”. Lines 54, 218, 236, 278, 320. 2. In the Introduction section, they must justify why they chose a biofilm model that used yeast and not filamentous phase to evaluate the antibiofilm activity of itraconazole, amphotericin B, and miltefosine. Considering that the filamentous phase is the predominant phase in the biofilms observed in vivo in cat's claw. The authors should consider using biofilm models using both morphological phases of the fungus (mycelium and yeast) to evaluate the anti-biofilm effect of the analyzed drugs. 3. The authors should explain why they did not consider including KI as a control, considered it the treatment of choice for Sporothrix infections. 4. Line 290: please indicate what was the predominant phase of the fungus in the observed biofilms. 5. Please explain in the Discussion section why In vitro Sporothrix spp. biofilms showed low susceptibility compared to miltefosine and amphotericin B; and also what antifungal mechanisms the authors may hypothesize could have miltefosine.

Author Response

Dear Reviewer 2,

We are grateful for all suggestions and the revision process that improved our manuscript. The manuscript “pathogens: 1553524” was revised according to the reviewer's comments.

Response to reviewer 2's comments are described below:

Reviewer #2:

This manuscript number pathogens-1553524 entitled: “Sporothrix spp. biofilms impact in the zoonotic transmission route: feline claws associated biofilms, itraconazole tolerance, and potential repurposing for miltefosine” by Pires dos Santos, et al., demonstrated the presence of Sporothrix fungal biofilms in the nails of cats diagnosed with sporotrichosis and evaluated the anti-biofilm effects of Sporothrix spp of drugs such as Itraconazole, amphotericin B, and miltefosine. The latter was suggested as an alternative to the use of itraconazole against biofilms of the fungus. These results support the use of miltefosine as a promising treatment alternative against sporotrichosis. Eventhough more studies should continue to be carried out for improving the effectiveness and safety in vivo in cats with sporotrichosis.

ANSWER: We thanked Reviewer 2 for the comments and changed the manuscript considering all suggestions.

The points to review in the document are the following:

  1. Modify Sporothrix sp by “Sporothrix spp”. Lines 54, 218, 236, 278, 320.

ANSWER: The modification was performed.

  1. In the Introduction section, they must justify why they chose a biofilm model that used yeast and not filamentous phase to evaluate the antibiofilm activity of itraconazole, amphotericin B, and miltefosine. Considering that the filamentous phase is the predominant phase in the biofilms observed in vivo in cat's claw. The authors should consider using biofilm models using both morphological phases of the fungus (mycelium and yeast) to evaluate the anti-biofilm effect of the analyzed drugs.

ANSWER: We include an explanation about the importance of using yeasts at starting inoculum in Sporothrix biofilm studies in the Introduction (lines 51-57). As described in the Discussion, since infected cats harbor mainly yeasts in their open wounds, the underlying characteristics of biofilms developed starting from yeasts can contribute to a better understanding of the pathophysiology of the disease (lines 213-215). 

  1. The authors should explain why they did not consider including KI as a control, considered it the treatment of choice for Sporothrix infections.

ANSWER: We did not include KI in the study because it does not inhibit Sporothrix growth in vitro at lower concentrations [doi: 10.1093/mmy/myy119 ].

  1. Line 290: please indicate what was the predominant phase of the fungus in the observed biofilms.

ANSWER: As described in the Discussion, we observed hyphae and yeast-like structures in biofilms present in claws from cats previously diagnosed with sporotrichosis (line 254).

  1. Please explain in the Discussion section why In vitro Sporothrix spp. biofilms showed low susceptibility compared to miltefosine and amphotericin B; and also what antifungal mechanisms the authors may hypothesize could have miltefosine.

ANSWER: The fact that concentrations up to 8 µg/ml of amphotericin B and miltefosine were able to inhibit Sporothrix biofilms is a good result, considering that several compounds inhibit planktonic fungal cells do not inhibit biofilms because biofilms were less susceptible to compounds due to cell density, extracellular matrix, and gene expression alterations [doi:10.5772/62768]. The mechanism of action related to the antifungal activity of miltefosine is not fully understood; however, it is partly due to the induction of apoptotic-like cell death and increased plasma membrane permeability [doi:10.1016/B978-0-12-809633-8.21026-1]. Spadari and coworkers (2018) and Rollin-Pinheiro and collaborators (2021) showed that miltefosine decreased mitochondrial membrane potential and increased ROS levels in Cryptococcus neoformans and Scedosporium aurantiacum [doi: 10.1128/AAC.00312-18, doi: 10.3389/fcimb.2021.698662]. Miltefosine induces pronounced disturbances in the plasma membrane in planktonic Sporothrix cells [doi: 10.1099/jmm.0.000041]. We hypothesized that miltefosine acts on Sporothrix biofilms by disrupting the cell membrane integrity, thus promoting an electrolyte imbalance and consequently the cell death of the fungus. This information was included in the Discussion (lines 263-278).